# Building XAI-Based Agents for IoT Systems

**Algirdas Dobrovolskis \***[ID]**, Egidijus Kazanavičius and Laura Kižauskienė**

Faculty of Informatics, Kaunas University of Technology, 44249 Kaunas, Lithuania
* Correspondence: algirdas.dobrovolskis@ktu.lt

**Abstract:** The technological maturity of AI solutions has been consistently increasing over the years, expanding its application scope and domains. Smart home systems have evolved to act as proactive assistants for their residents, autonomously detecting behavioral patterns, inferring needs, and making decisions pertaining to the management and control of various home subsystems. The implementation of explainable AI (XAI) solutions in this challenging domain can improve user experience and trust by providing clear and understandable explanations of the system's behavior. The article discusses the increasing importance of explainable artificial intelligence (XAI) in smart home systems, which are becoming progressively smarter and more accessible to end-users, and presents an agent-based approach for developing explainable Internet of things (IoT) systems and an experiment conducted at the Centre of Real Time Computer Systems at the Kaunas University of Technology. The proposed method was adapted to build an explainable, rule-based smart home system for controlling light, heating, and ventilation. The results of this study serve as a demonstration of the feasibility and effectiveness of the proposed theoretical approach in real-world scenarios.

**Keywords:** Internet of things (IoT); smart house; multi-agent systems; explainable artificial intelligence (XAI)

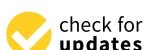



## 1. Introduction

The technological maturity of AI solutions is undeniably increasing year by year, widening its application scope and domains. Technology-enriched domestic environment, which has always been at the heart of AI research, is getting progressively smarter and more accessible to its end users. Contemporary smart home systems act as proactive assistants to their residents, autonomously detecting users' behavioral patterns, inferring their needs, and making decisions regarding the management and control of various subsystems. With the growing reliance on sophisticated AI systems in daily lives, it is becoming increasingly important to be able to understand and explain the decisions made by AI models, as they directly or indirectly affect many aspects of our lives, such as human well-being, safety, comfort, energy efficiency, etc. The reliability of AI solutions is one of the most important factors for the continuity of their adoption, and it can be reinforced by enhancing AI models with explanatory features. This is where the demand for explainable artificial intelligence (XAI) comes in—a research area that is gaining momentum in the field of machine learning and AI applications. XAI is the set of techniques and methods to convert the so-called black-box AI algorithms to white-box algorithms, where the results achieved by these algorithms and the variables, parameters, and steps taken by the algorithm to reach the obtained results, are transparent and explainable [1]. The most-cited benefits of XAI are improved decision-making, increased trust and transparency, improved user satisfaction, debugging and model improvement, and better compliance with regulations [2–6].

### 1.1. The Need for Explainability in Smart Home Domain

Some AI and ML application domains can be considered very sensitive and high-risk, closely associated with human lives, their wellness, and safety—industrial control,

healthcare, military domain, or self-driving cars perfectly represent instances of such safety-critical domains. Most authors agree that the demand for XAI in safety-critical domains is inseparable from further developments, as they involve high-consequence decisions, and a failure of the AI system could result in significant harm to people or the environment [7–9].

Smart homes are not generally considered a safety-critical domain because the consequences of the AI system's failure in a smart home setting are typically less severe than in other safety-critical domains such as transportation or healthcare. However, smart homes are challenging environments for XAI, as they are decentralized systems that undergo runtime changes. Moreover, an explanation is more likely to be required when the user is faced with a contradicting situation [10]. There are some scenarios where the failure of the AI system in a smart home could potentially lead to harm. For example, if the system controls heating, lighting, or other safety-related functions, a failure could result in unsafe conditions. Thus, it is important that AI-based systems, especially for home security or controlling higher-risk devices, provide clear explanations for their decisions and potential outcomes. The actions of various AI-enabled gadgets, minimizing the effort and time required for everyday household activities, should also be transparently explained, especially if they do not fully meet users' preferences. Overall, smart home systems should take responsibility not only for real-time control behavior but for protective behavior as well if potential threats have been recognized for the security of humans, their personal data, or assets. Unintended consequences, system malfunctioning, false notifications of various home subsystems, and unpredicted errors are key concerns undermining trust and confidence in any AI solution. XAI is important in smart home settings because it can help users understand and interact with the AI system in a more natural and intuitive way. By providing explanations for its decisions, the AI system can increase users' trust and understanding of the system, which can lead to greater acceptance and adoption.

The research on the application of XAI to the smart home domain focuses on various aspects. A study in [11] applied XAI principles to the domain of activity recognition in smart homes, aiming to provide end users with transparent explanations for the system's decision-making processes. The main idea behind the research is to propose and evaluate four computational techniques for generating natural language explanations of smart home data based on explainable artificial intelligence (XAI) methods. The paper evaluates the effectiveness of each technique in generating accurate and useful explanations and presents recommendations for the best approach in the domain of smart home automation. Another study in [12] provides valuable insights into the current state and future potential of XAI in the power system domain, reviews XAI techniques for energy and power system applications in smart buildings, and helps in understanding the associated challenges and potential research directions in this field. In [13], the authors propose a modular architecture for self-explanatory smart homes, addressing the challenges of heterogeneous devices, runtime changes, and varying contexts.

### 1.2. Explainable Multi-Agent Systems for IoT

Our choice to model IoT systems using an agent-based approach is based on several important considerations. First of all, the concept of an "agent" has been used as a metaphor for defining autonomous entities of AI systems for more than two decades, representing both humans and computers, physical robots, and virtual assistants. The authors in [14] agree that the general notion of an agent is so wide that both software entities and human beings may fit it and that such formal laxity is deliberate and useful because it allows human-machine and machine-machine interactions to be captured at the same level of abstraction. Moreover, a recent survey on agent-based Internet of things [15] states that the use of agents in IoT systems is motivated by their ability to model and simulate complex, dynamic, and autonomous systems. In the context of IoT, agents can represent devices, services, or applications and can cooperate to achieve a common goal, such as efficient resource management or real-time data analysis. Furthermore, the agents' programming paradigm allows for the encapsulation of control and coordination based on

high-level asynchronous message-passing mechanisms, which can facilitate interoperability, scalability, and adaptability in the development of IoT systems.

Though the research area of explainable agent-based systems is a relatively new and emerging field within the broader area of artificial intelligence (AI), the number of related studies is gradually increasing, and they are being carried out in different directions. For example, recently, there have been increasing efforts to explain ML models using virtual [16,17] or conversational agents [18], which are closest to people in their way of communicating and, therefore, more acceptable to people. Experiments have shown that XAI principles implemented through conversational and virtual agents contribute to increasing user trust in the XAI system [17]. The authors in [9] discuss the concept of real-time multi-agent systems and how they are designed to operate in highly dynamic environments, highlighting the importance of meeting both soft and hard deadlines. The paper also explores the use of BDI-based agents, which are suited for unpredictable scenarios requiring dynamic decision-making, and how they can be used to develop explainable and real-time compliant MAS. Additionally, the research to date on explainable multi-agent systems comes under various definitions, namely "XMAS" [9], "XMASE" [19], "Explainable agents" [20], "Explainable Agency" [21], or goal-driven XAI [20].

To sum up, recent studies advocated that multi-agent systems offer a coherent yet expressive set of abstractions, promoting conceptual integrity in the engineering of complex software systems and serving the purpose of social XAI [14]. Furthermore, the growing body of research supports the idea that multi-agent systems can be beneficial for explainable AI (XAI) by providing improved decision-making, increased trust and transparency, and better compliance with regulations [22–24].

The main objective of this paper is twofold: to propose a method for building explainable agent-based IoT systems and to develop an application for controlling a smart home system in a real-time environment. This article addresses the problem of building explainable IoT systems that can control smart home environments in real-time using agent-based architectures. Specifically, this article proposes a method for developing these systems that incorporate explainable AI, which is important for ensuring transparency and user trust in IoT systems.

## 2. Agent-Based Method for Building Explainable IoT Systems

IoT systems are designed and implemented as a group of autonomous, collaborative components, each having dedicated sensing, actuation, processing, or control functions. It is a collection of various hardware and software agents. Although there are quite a lot of IoT architectures proposed recently in [25–27], they have no integrated means for explainability included. The following method complements existing IoT architecture models by integrating an explanation interface.

### 2.1. Reference Architecture for an Agent-Based IoT System

Figure 1 illustrates the reference architecture of an agent-based Internet of things (IoT) system, which consists of two main components: the virtual and physical environments. The system is composed of software agents, designated with the letter A, that operate in the virtual environment and interact with physical components, such as sensors and actuators, in the physical environment. Physical sensors, which are deployed in the field to collect data on parameters such as temperature, presence, and humidity, can range in complexity from simple, reactive data collection and transmission devices to more advanced devices with onboard processing and storage capabilities. Actuators, on the other hand, are used to perform physical actions, such as opening or closing valves, turning motors on or off, or activating lights. Each physical sensor and actuator is associated with corresponding sensors and actuator agents in the virtual environment. Real-time data from the physical sensors are constantly acquired by sensor agents and passed to data collection agents, which act as intermediaries between the lowest-level sensor/actuator agents and higher-level decision support agents. Data collection agents pre-process the data and, when certain

predetermined thresholds are exceeded, forward the information to decision support agents for further analysis and inference.

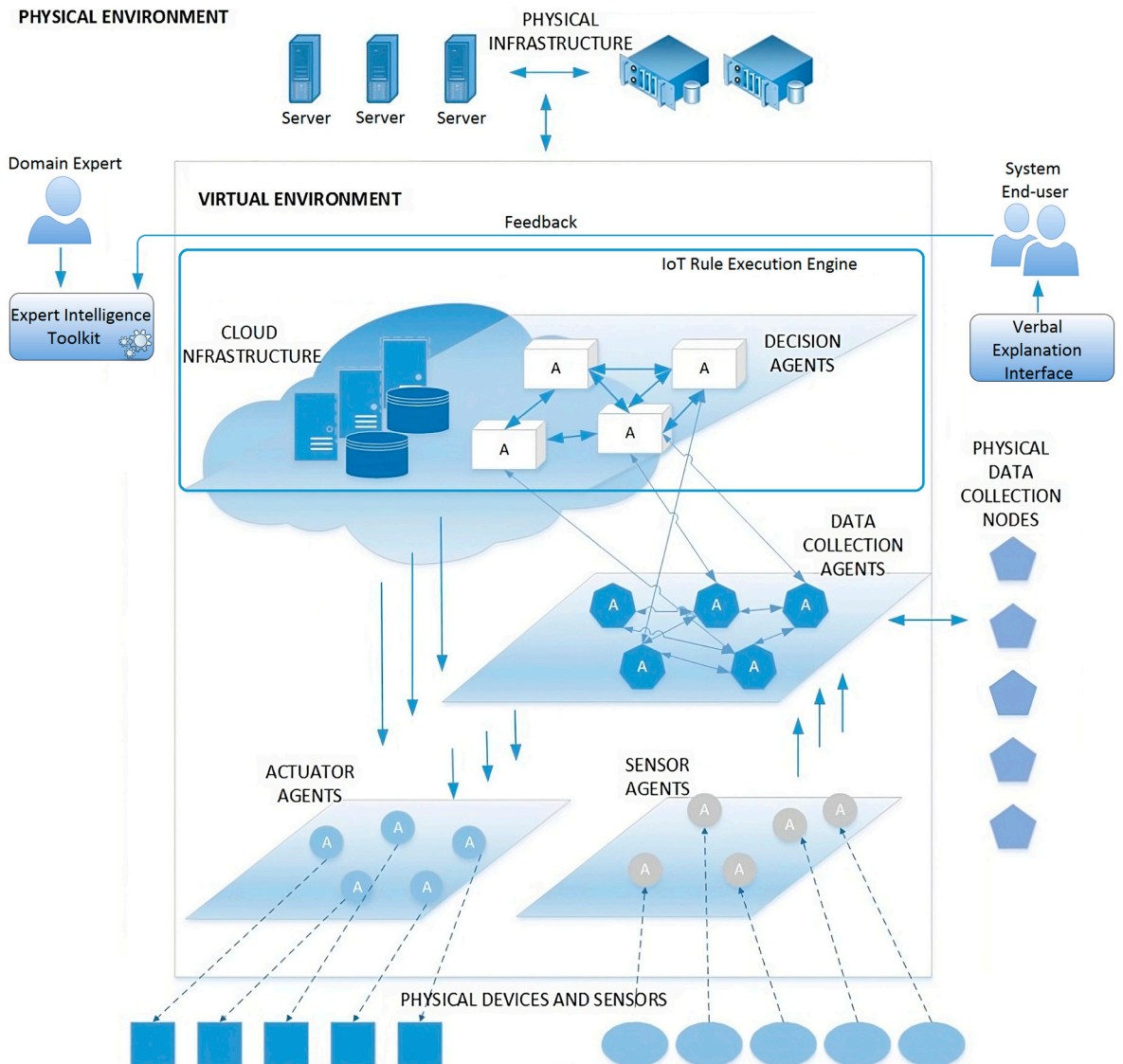

**Figure 1.** Agent-based IoT system architecture.

The users, depicted in Figure 1, represent the domain expert and end users of the system. The domain expert, who is typically the developer of the system, is responsible for defining the system's behavior rules, providing the necessary data, and specifying user preferences. This is accomplished using the Expert Intelligence Toolkit, a component used to convert knowledge into control rules for the IoT rule execution engine. The IoT rule execution engine is a major component of the system and is based on underlying decision agents. It is responsible for collecting data from sensors, processing the data, and transmitting control settings to the actuators based on the established control rules. Additionally, the rule execution engine can perform other functions, such as device management and security. It can be hosted on a public cloud service such as Amazon Web Services or on a private server located on-premises. The rule execution engine typically includes a database for data storage and tools for data visualization. The explanation interface is software used for user interaction with the system. It provides the necessary information and explanations of the system's decisions to the end users, allowing them to understand the actions taken by the system. A web-based interface is recommended to maximize accessibility through a variety of user-owned devices such as computers, tablets, and smartphones. After receiving

the verbal explanation, users can provide feedback through the Expert Intelligence Toolkit, allowing the system to adapt to their current needs.

The denoted system components are connected through a network, which can be both wired and wireless. The network infrastructure will depend on the requirements of the IoT system, such as the type and amount of data being transmitted, the required reliability and latency, and the availability of a network infrastructure in the deployment area. Communication messages are transmitted through MQTT (Message Queuing Telemetry Transport)—a lightweight publish/subscribe messaging protocol that is commonly used for IoT applications because of its low overhead and good support for intermittent and low-bandwidth networks.

Compared to similar approaches which use services or microservices, the proposed method stands out in several ways:

Explainability: the method focuses on building agent-based IoT systems that are explainable, meaning that the system's behavior can be understood and traced back to specific actions of individual agents.

Flexibility: the proposed method is designed to be flexible and adaptable to different use cases and scenarios, allowing for the development of a wide range of IoT applications.

Real-time performance: the developed application for controlling a smart home system can operate in real-time, with fast response times and low latency.

Integration with existing technologies: the method can be integrated with existing technologies and frameworks using MQTT as the communication interface, making it easier for developers to adopt and use.

### 2.2. IoT System Implementation Method

The proposed agent-based IoT system implementation method is provided in Figure 2. The process covers 5 basic steps, from defining control rules to testing the explanation interface output.

Step 1. Define the control rules for the rule execution engine using Expert Intelligence Toolkit. Already available solutions such as Matlab's Fuzzy Toolkit can be used for the task as well as custom in-house solutions. Adding machine learning techniques can be used to improve the decision-making capabilities of the underlying decision agents. This could involve training models on historical data and using them to predict future states and recommend actions.

Step 2. Apply a generated control code (script) to the rule execution engine.

Assign each agent an MQTT subscription channel to receive commands and send status to the rule execution engine.

Step 3. Test the communication, i.e., whether sensor agents send MQTT messages, react to data changes, and whether actuator agents respond to commands received through MQTT.

As IoT systems often involve sensitive data, it is important to consider privacy and security implications when designing explainable agent-based systems. This could involve encrypting communications, implementing access controls, and anonymizing data where possible.

Step 4. Integrate and test if the rule execution engine functions properly according to the control code—underlying decision agents should make decisions depending on the acquired sensor data and send corresponding control signals to actuators.

Depending on the complexity of the IoT system and the number of agents involved, real-time performance could be a concern. To address this, techniques such as distributed computing, parallel processing, and edge computing could be used to improve the speed and efficiency of the system.

Step 5. Output logs of the rule execution engine are converted to verbal explanations using the Explanation Interface. Analyze user feedback and adjust rules if user preferences have changed. To improve the performance and accuracy of the system over time, an automated process for tuning the control rules should be implemented. This in-

volves monitoring the system's performance and adjusting the rules as needed to improve its effectiveness.

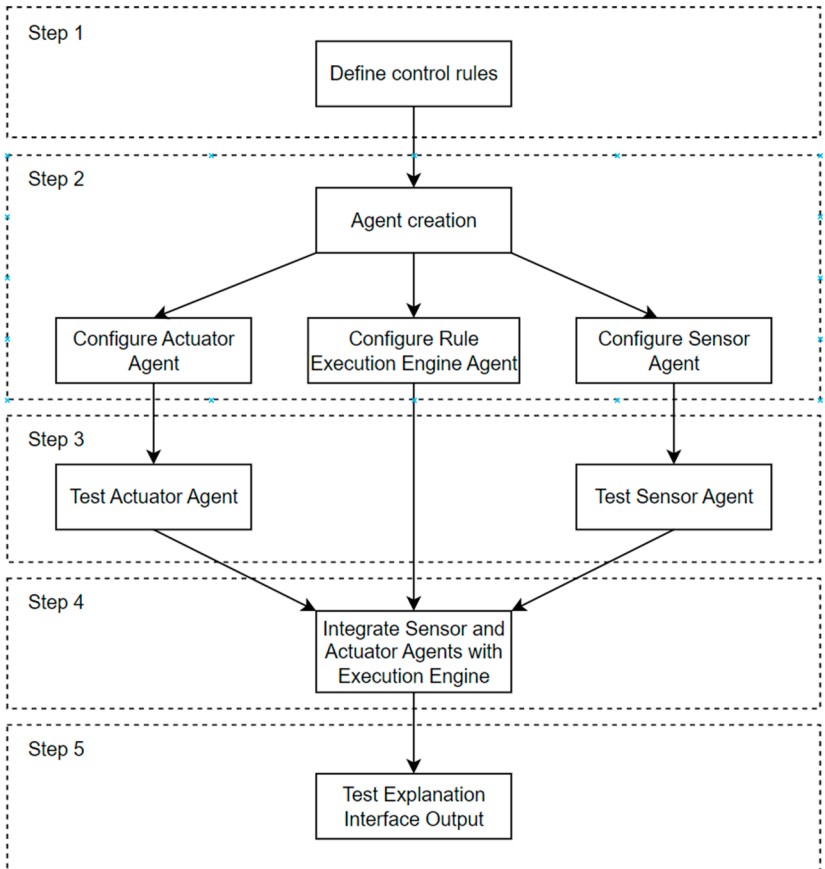

**Figure 2.** Method steps.

## 3. Results—An Explainable Agent-Based Smart Home System

The proposed method was adapted to a real-world scenario for controlling light, heating, and ventilation in a smart home setting. The conducted experiment allowed for testing of the proposed theoretical approach in a real-time IoT environment.

### 3.1. Experimental Setup

The following hardware equipment was required for the experiment:

(a)  HP Pavilion all-in-one—24 computer—as IoT rule execution engine;
(b)  SONOFF T3 TX Series WIFI Wall Switch—light switches;
(c)  DANFOSS, TWA-K 24V, M30X1.5, NC—thermal actuators;
(d)  Asus wireless router for transmitting data;
(e)  SONOFF R2 4 Channel—relay block for switching lighting and heating;
(f)  SONOFF® RF Bridge 433 MHz—for ventilation control;
(g)  SONOFF® PIR2 Wireless Infrared Detector—for presence detection;
(h)  ESP32 microcontroller with DS18B20 temperature sensor—for indoor temperature measurements;
(i)  M5 Stack microcontroller with light sensor.

Every hardware device is treated as a hardware agent. All hardware agents are communicating through the MQTT protocol sent through an ASUS router. The deployment and the connections between hardware agents are shown in Figure 3.

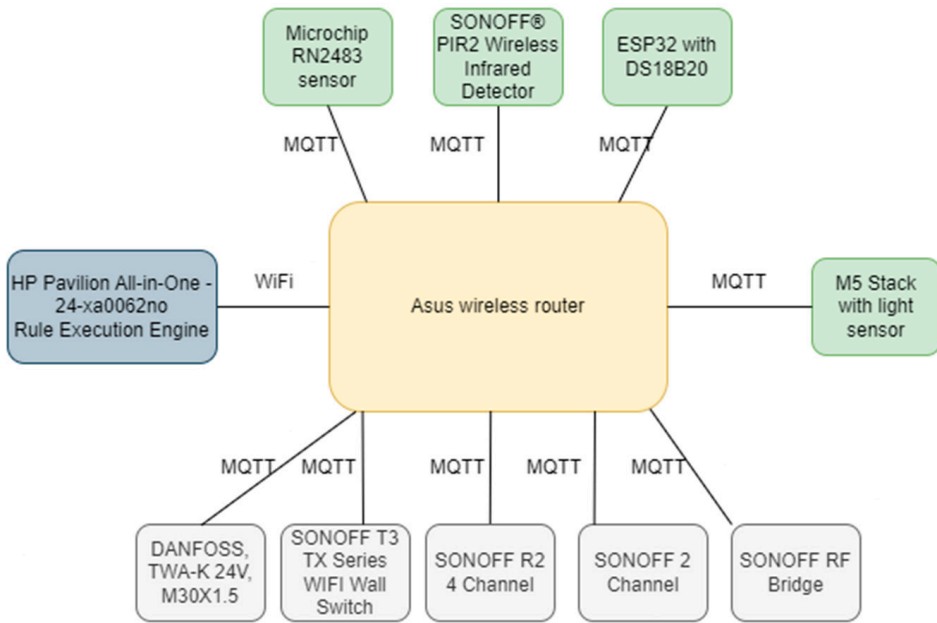

**Figure 3.** Communication of hardware agents.

The software components, required for the experiment are enumerated below:

i.     Ubuntu20.04 Linux;
ii.    SQLite database;
iii.   Node.js 16 framework;
iv.    Pimatic—smart house control environment.

### 3.2. The Steps for Developing an Explainable Smart Home Control System

The proposed method has been adapted to implement a smart home control system. The experiment was carried out according to the steps outlined in Figure 2. *Matlab's Fuzzy Toolkit* was used to define the control rules for **step 1** of the proposed method. For this purpose, the model was divided into 3 parts to represent the different controllers used in home automation: heating, ventilation, and lighting. The heating model with input data [temperature] and [presence] is shown in Figure 4. The heating model utilizes two inputs, temperature and presence, and one output, heating level. The [temperature] input is described by a membership function with verbal values of {very cold, cold, warm, hot}. The membership function is represented by Gaussian functions with temperature values ranging from 0 to 25 degrees Celsius. The [presence] input is described by a membership function with verbal values of {not present, present}. The output, heating level, ranges from 0, indicating heating is off, to 3, indicating high-level heating.

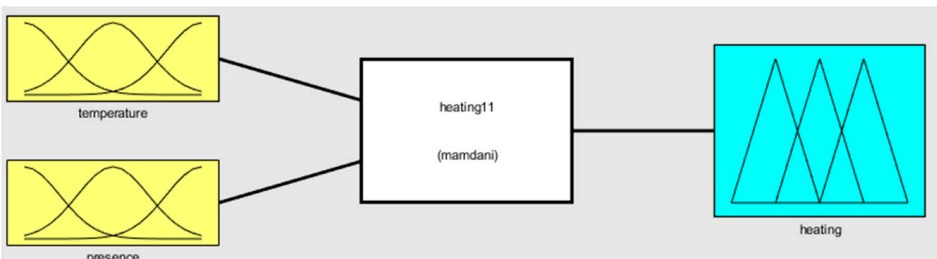

**Figure 4.** Fuzzy heating model.

The lighting model also utilizes two inputs, lighting and presence, and one output, lighting level. The [lighting] input is described by a membership function with verbal values of {very dark, dark, light, very light}. The membership function is represented

by Gaussian functions with light values in lumens ranging from 0 to 3000. The [presence] input is described by a membership function with verbal values of {not present, present}. The output, lighting level, ranges from 0, indicating lighting is off, to 3, indicating high-level lighting.

Finally, the ventilation model utilizes two inputs, temperature and presence, and one output, ventilation level. The [temperature] input is described by a membership function with verbal values of {cold, warm, hot, very hot}. The membership function is represented by Gaussian functions with temperature values ranging from 0 to 45 degrees Celsius. The [presence] input is described by a membership function with verbal values of {not present, present}. The output, ventilation level, ranges from 0, indicating ventilation is off, to 5, indicating high-level ventilation.

To verify if the heating control rules covered all cases of control scenarios, the graph relationship between the heating level with temperature and presence is demonstrated in Figure 5. It was determined that when a person is present, the heating will gradually increase as the temperature inside decreases, with low heating activated when the room is warm, medium heating when it is cold, and high heating when the room temperature is very cold. When the person is absent, the system will maintain minimal heating to prevent damage to the room from cold and humidity. It was established that all scenarios had been accounted for, ensuring proper operation of the heating agent.

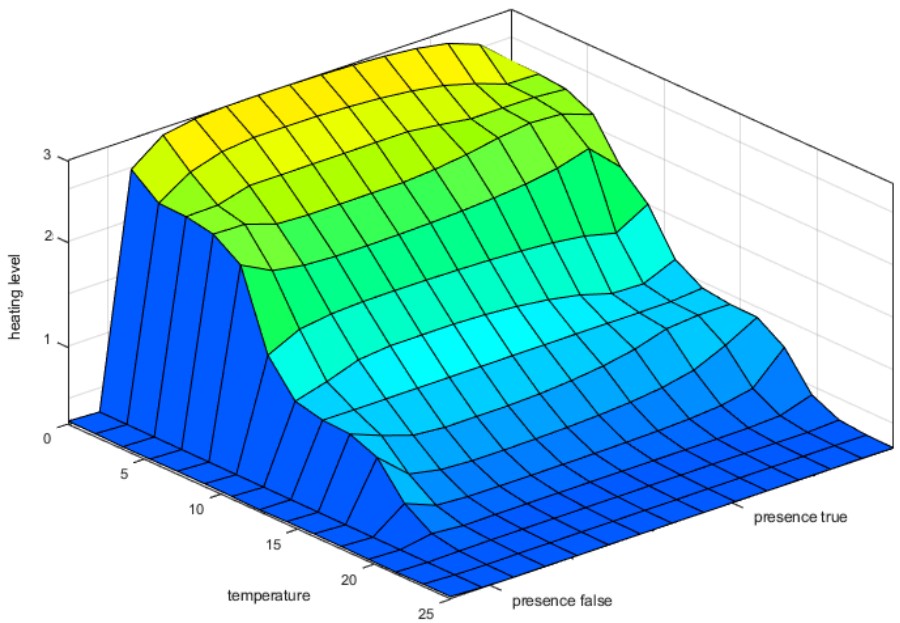

**Figure 5.** Heating level dependency on temperature and presence.

The correlation between lighting level and luminosity within the room as well as presence is depicted in Figure 6. Through verification, it was determined that when an individual is present, the lighting level will gradually increase as the luminosity decreases, with 20% brightness activated when the light is low, 50% brightness activated when it is dark, and 100% brightness activated when the room is extremely dark. Conversely, when the individual is absent, the system will shut off the lighting to minimize energy consumption. It has been confirmed that all possible scenarios have been taken into consideration, thus ensuring the optimal functionality of the lighting agent.

The relationship between ventilation level and temperature, as well as presence, is illustrated in Figure 7. Through examination, it was established that when an individual is present, the ventilation level will be adjusted proportionally to the increase in temperature within the room. Specifically, when the temperature is warm, level one ventilation is activated; when it is hot, level three ventilation is activated; and when it is extremely hot, level five ventilation is activated. Conversely, when the individual is absent, the system

will maintain minimal ventilation to preserve the well-being of any plants present. It has been confirmed that all potential scenarios have been considered, thereby ensuring the optimal performance of the ventilation agent.

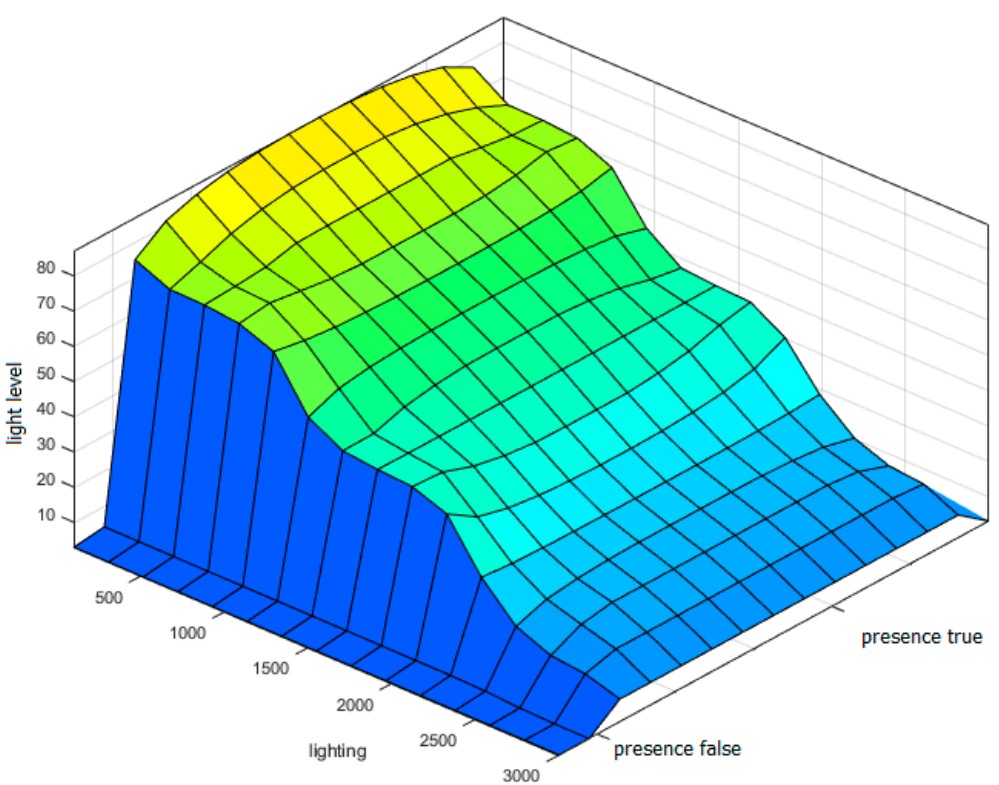

**Figure 6.** Lighting level dependency on presence and luminosity of light.

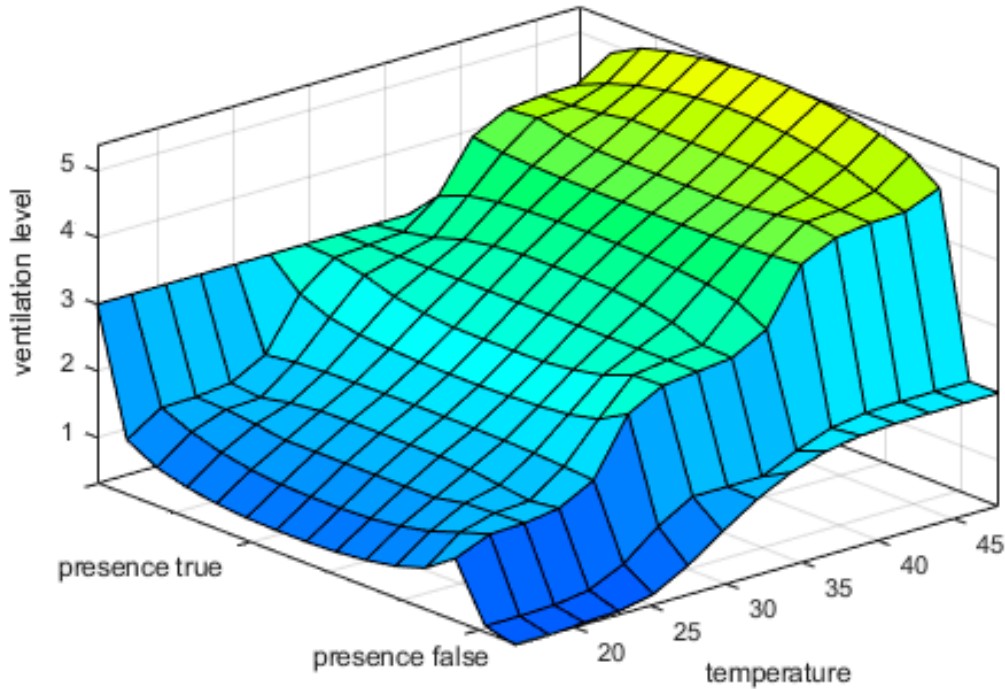

**Figure 7.** Ventilation level dependency on temperature and presence.

Following the second step of the proposed method, the Pimatic home automation environment was installed and configured on an HP Pavilion all-in-one computer utilizing

the Ubuntu operating system. Additionally, sensor and actuator agents were loaded with custom firmware to facilitate the transmission and receipt of MQTT messages. The following step involved the conversion of the generated verbal rules, as depicted in Figure 8, into control software rules. Specifically, for the heating system, six rules were established for when the user is present, and one rule was implemented to maintain a minimal temperature when the user is absent and the temperature is low.

```
1. If (input1 is very_cold) and (input2 is present) then (output1 is high_heating) (1)
2. If (input1 is cold) and (input2 is present) then (output1 is medium_heating) (1)
3. If (input1 is warm) and (input2 is present) then (output1 is low_heating) (1)
4. If (input1 is cold) and (input2 is not_present) then (output1 is low_heating) (1)
5. If (input1 is warm) and (input2 is not_present) then (output1 is low_heating) (1)
6. If (input1 is hot) and (input2 is not_present) then (output1 is low_heating) (1)
7. If (input1 is very_cold) and (input2 is not_present) then (output1 is low_heating) (1)
```

**Figure 8.** Verbal control rules for heating.

In our smart home automation system, the ventilation system has been programmed with seven rules for when the user is present and one rule for when the user is absent to maintain minimal ventilation when the temperature is extremely high. Additionally, the lighting system is activated solely when an individual is present in the room, with different levels being defined by five rules. These rules were established based on the analysis of temperature, luminosity, and presence data collected from the smart home environment.

In the third step of the experiment, the sensor and actuator agents were tested using MQTTBox software to evaluate their ability to transmit data and receive commands accurately. The Pimatic home automation environment, which features a graphical rule definition interface, eliminated the possibility of syntax errors, and only devices that were present within the environment could be defined.

In the fourth step of the experiment, the sensor and actuator agents were incorporated into the Pimatic environment and were tested using the Pimatic GUI to display data. The functionality of the smart home environment was evaluated through the Pimatic user interface, as depicted in Figure 9.

In the fifth step of the experiment, user notifications were generated in the event of any modifications in the control environment. The relationship between the invoked rule and the action taken was implemented using an eFLL library. The notifications consisted of a text message displaying the time, the type of message (in this case, "info" denoting a user notification), the name of the rule, the room number and sensor status that was activated, and the agent device that received the command of "on" or "off". After testing, certain values for the lighting system were adjusted as the sensor in room 314 was providing lower readings, and the light was not functioning optimally in terms of turning on and off. The explanation output of the system is presented in Figure 10.

In the current case, a verbal explanation approach was used to provide information for users on why a certain action was taken based on the activated rule from the rule set, as opposed to numerical explainability, where the explanation is expressed in mathematical or statistical terms, such as feature importance scores or model coefficients. Verbal explainability is often preferred in situations where there is a need for a human to be able to interpret and understand the decision-making, losing statistical evaluation ability in the process.

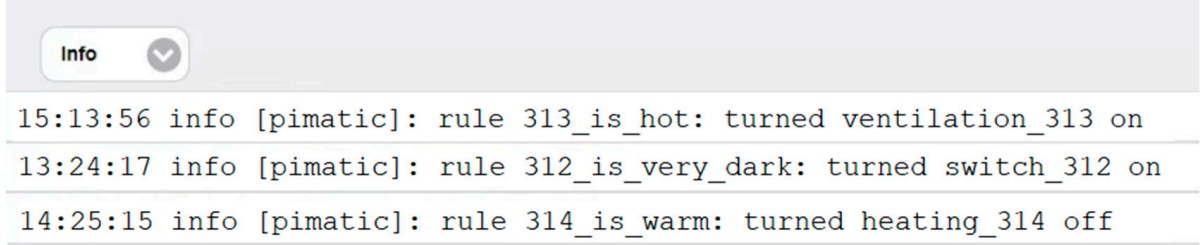

**Figure 9.** Pimatic user interface controlling house environment.

```
Info  ⌄

15:13:56 info [pimatic]: rule 313_is_hot: turned ventilation_313 on
13:24:17 info [pimatic]: rule 312_is_very_dark: turned switch_312 on
14:25:15 info [pimatic]: rule 314_is_warm: turned heating_314 off
```

**Figure 10.** Examples explaining system behavior.

## 4. Conclusions and Future Work

The study presented in this paper aimed to address the issue of lack of transparency in smart home systems, which often leads to decreased trust and engagement among users. To address this issue, an explainable agent-based approach was proposed for developing IoT systems, and an experimental study was conducted in real-life application scenarios at the Centre of Real Time Computer Systems at the Kaunas University of Technology. The study was conducted over the course of one year in three laboratory rooms, providing a comprehensive examination of the system's functionality during the four seasons. The results of the study have demonstrated that the proposed explainable agent-based approach can increase trust and engagement among the users and enable a better understanding and control of the system. The developed application demonstrates the practical application of this technology in controlling a smart home system. With the ability to monitor and control

their home remotely in real-time, users can increase their efficiency and security while also reducing their energy consumption.

The proposed method for building explainable agent-based IoT systems can be adapted and used in various other scenarios beyond the control of smart home systems. For instance, the same method can be applied in the field of healthcare, where IoT devices are used to monitor and control medical processes. The agent-based IoT system can be designed to provide explainable decision-making capabilities to the medical processes, allowing doctors and other healthcare professionals to understand how decisions are made and the reasons behind them. This can help in improving patient outcomes, reducing medical errors, and increasing patient trust in the system.

However, there are also potential challenges and limitations to consider. The development of these agent-based IoT systems requires significant resources and expertise, which may limit their accessibility to smaller organizations or individuals. Additionally, ensuring the security and privacy of these systems will be essential to prevent unauthorized access or misuse of sensitive data.

Regulations such as the General Data Protection Regulation (GDPR) and other data privacy laws require that automated decision-making systems must be transparent and explainable to the individuals affected by the decisions. Therefore, implementing XAI can help ensure compliance with these regulations and increase trust in automated decision-making systems, also leading to faster adoption.

Overall, the proposed method for building explainable agent-based IoT systems can be applied in any scenario where IoT devices are used to monitor and control processes and where the ability to provide clear explanations for decision-making is crucial.

Future research aims at automating code generation and implementing an interface to use verbal explanations in the other case studies while focusing on the performance, effectiveness, and security of these systems and identifying areas for improvement and new opportunities for development.

**Author Contributions:** Conceptualization, E.K.; Software, A.D.; Formal analysis, L.K.; Investigation, A.D.; Writing–original draft, A.D.; Writing–review & editing, L.K.; Supervision, E.K. All authors have read and agreed to the published version of the manuscript.

**Funding:** This research received no external funding.

**Institutional Review Board Statement:** Not applicable.

**Informed Consent Statement:** Not applicable.

**Data Availability Statement:** Data sharing not applicable.

**Conflicts of Interest:** The authors declare no conflict of interest.

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
