# Peer review of "Building XAI-Based Agents for IoT Systems"

_applsci, doi:10.3390/app13064040_

Round 1
Reviewer 1 Report
The paper proposes an approach to built IoT systems using explainable agents, with a case study in Kaunas University.
Although the topic and problem covered by the paper is interesting and relevant to scientific community, the authors missed the opportunity to clearly show their contributions considering state of the art. The paper needs to be substantially reorganised and detailed, including more evaluation to be considered ready for publication. Only if the following aspects are successfully addressed the paper could be considered at a later stage.
Main aspects to address:
- Need to make clear the contribution of the architecture proposed compared to other many existing IoT architectures. For this authors need to incorporated a clearly identified literature review/related work section and highlight the key gap they are trying to address.
- The agent notion used is too broad and it is not clear what are the characteristics of agents that are exploited in the method, I.e. autonomy? Authors need to clarify within the paper: What decisions every type of agent (sensor, actuator, rule execution) are making? What differentiates these “agents” from standard services or micro services widely used to build IoT systems? What are the characteristics of these "agents"? what are their capabilities and "intelligence"? What are they trying to explain (from the "explainaible AI" perspective)?
- The proposed method seems standard to building an IoT application and only step 5 at the end is added for support of explainability: “Test Explanation Interface“ but there are no details about what this interface is and how this is build. This could be a more interesting step to discuss and contribute with but authors not provide any detail about this in the paper.
- Evaluation needs to be significantly improved comparing against state of the art, highlighting the explainability features of the solution proposed and where it standout compared to existing approaches.
- A discussion section needs to be added indicating the strengths, weaknesses of the method proposed, lessons learned and especially, how this method can be used beyond the case study introduced.
- Figure 1 quality is really poor, seems to be taken from elsewhere, e.g. there a some servers at the top left of the figure with some text below but the text is not readable at all, other text is barely readable. Pease improve quality of this figure.
- Comments about Figure 5-7 are mainly descriptive but no clear what is the point of these figures in supporting the use of the method proposed and its benefits. What do these figure have to do with the explainability of the system?
These changes require time and authors are advised to take good time to address these comments properly instead of rushing a quick response. Unfortunately if changes are not addressed properly the paper won't be progressed.
Author Response
Thank You for the freedback, changes were made according to comments nubered by order:
1. improved introduction, stating differences from simmilar methods
2. narrowed notion of agents, stating the main aspects of using them in IoT: to act autonomously, communicate with each other in a decentralized manner, and maintain some degree of internal state, allowing them to remember previous actions and make decisions based on that information.
3. Difference from standart IoT systems is the use of agents, which adds explanaition interface is a subsystem which resolves what fuzzy rule was invoked and action was taken, the process of building it was achieved with the use of eFLL library, which is now mentioned in section 3.2
4. section 2.1 was improved describing what stands out from simmilar approaches.
5.discussion section was improved adding section about strengths and weakness of proposed method.
6. improved figure 1 quality and enlarged text to be more readable
7. figures 5 to 7 are displayed for control rule evaluation, demonstrating its effectiveness in building explainable and efficient IoT systems.
grammar and punctuantion errors in the article were corrected
Reviewer 2 Report
The figures should be clearer
(199) Figure 1. Agent-based IoT system architecture
(Figure 9. Pimatic user interface controlling house environment.
(351)Figure 9. Pimatic user interface controlling house environment.
Author Response
improved english grammar and punctuation, improved image quality in figures 1 and 9
Reviewer 3 Report
The paper presents an agent-based approach for developing explainable IoT systems, and an experiment is also presented. The results of this study serve as a demonstration of the feasibility and effectiveness of the proposed theoretical approach in real-world scenarios. The topic is valuable, but some points should be improved.
1. The Abstract should state the principal objectives and scope of the investigation. For instance, why it is necessary to build explainable agents for IoT systems. Generally, explainable agents are essential when used in contexts where decisions have substantial implications for those affected and where there is a requirement for legal compliance. I suggest describing the motivation more clearly in the Abstract.
2. The Introduction has too many contents; parts of this section can be organized as a new section. In addition, the background and definition of the problems need to be better defined, and the authors need to relate the specific background information to the main purpose of the present paper. It needs to be clarified what problem they are addressing and why they chose the proposed methodology.
3. The workload of your work is sufficient, but it's recommended to emphasize your innovation. In addition, how to define the explainability of an agent?
4. In section 2, you should discuss the experimental results further; please note that the research article differs from an experimental report.
5. The Conclusion section is too weak; please note that the Conclusion is not the replication of the Introduction; the authors need to summarize the evidence for each Conclusion and present the generalizations the results show.
Author Response
Thank you for Your improvement suggestions, article was edited with:
- abstract was modified with text: "The article discusses the increasing importance of explainable artificial intelligence (XAI) in smart home systems, which are becoming progressively smarter and more accessible to end-users and presents an agent-based approach for developing explainable Internet of Things (IoT) systems and an experiment, conducted at the Centre of Real Time Computer Systems at Kaunas University of Technology. The proposed method was adapted to build a explainable, rule based smart home system for controlling light, heating, and ventilation. "
Since addressing legal compliance in IoT systems requires a multidisciplinary approach that involves legal experts, policymakers, and technical experts working together to develop policies, regulations, and technical solutions it was not addressed in the article but was added in discussion section of the article.
2. introduction was shortened and agent based explainable method for IoT was moved to a separate chapter.
3. innovation was emphasized by adding a chapter about main features of agents used in proposed method, discussion about verbal explanation and added description of explanation interface implementation.
4. added discusion of experiment results in section 3 (was section 2 before)
5. conclusion was updated with result generalisation and future works.
grammar and punctuantion errors in the article were corrected
Round 2
Reviewer 1 Report
The fundamental issues addressed in the first review were not addressed.
Author Response
changed references to more relevant ones [3, 7], added extra references that are releveant to research [24-28]
defined research objective for proposed method
improved description of proposed method with 3 key differences from similar proposals: explainability, real-time performance and flexibility.
improved experiment descriptions
improved results and future works section
added formatting, grammar and punctuation improvments
Reviewer 3 Report
The authors have addressed my concerns.
Author Response
changed references to more relevant ones [3, 7], added extra references that are releveant to research [24-28]
improved formatting, grammar and punctiation